ecology, biological applications, ecosystems

global change, functional traits, functional biogeography, environmental filter, ecological prediction, niche

**Author for correspondence:**
Stephanie J. Green
e-mail: stephanie.green@ualberta.ca

# Trait-based approaches to global change ecology: moving from description to prediction

Stephanie J. Green[1], Cole B. Brookson[1], Natasha A. Hardy[1,2] and Larry B. Crowder[2]

[1]Department of Biological Sciences, University of Alberta, Edmonton, AB, Canada
[2]Hopkins Marine Station of Stanford University, Pacific Grove, CA 93950, USA

(iD) SJG, 0000-0003-4705-7859; CBB, 0000-0003-1237-4096; NAH, 0000-0002-9003-6204;
LBC, 0000-0003-3131-2579

Trait-based approaches are increasingly recognized as a tool for understanding ecosystem re-assembly and function under intensifying global change. Here we synthesize trait-based research globally ($n = 865$ studies) to examine the contexts in which traits may be used for global change prediction. We find that exponential growth in the field over the last decade remains dominated by descriptive studies of terrestrial plant morphology, highlighting significant opportunities to expand trait-based thinking across systems and taxa. Very few studies (less than 3%) focus on predicting ecological effects of global change, mostly in the past 5 years and via singular traits that mediate abiotic limits on species distribution. Beyond organism size (the most examined trait), we identify over 2500 other morphological, physiological, behavioural and life-history traits known to mediate environmental filters of species' range and abundance as candidates for future predictive global change work. Though uncommon, spatially explicit process models—which mechanistically link traits to changes in organism distributions and abundance—are among the most promising frameworks for holistic global change prediction at scales relevant for conservation decision-making. Further progress towards trait-based forecasting requires addressing persistent barriers including (1) matching scales of multivariate trait and environment data to focal processes disrupted by global change, and (2) propagating variation in trait and environmental parameters throughout process model functions using simulation.

## 1. Trait-based environmental filters of distribution and abundance

Identifying general principles that govern the distribution and abundance of species across Earth's ecosystems is a fundamental pursuit in ecology. Over the past three decades, trait-based ecology (or 'functional biogeography')—focused on the role that measurable organism characteristics play in mediating geographic distribution and abundance—has emerged as a major conceptual lens through which to describe general processes driving patterns of biodiversity across the biosphere [1]. Trait-based theory stemmed from correlations between the frequency of phenotypic traits hypothesized to affect individual and population-level success across taxa and environmental contexts (i.e. functional traits [2]). As bodies of evidence identifying key traits that influence ecological processes (i.e. mechanisms) grew, scientists acquired the means to predict ecological outcomes in new contexts (such as following disturbance) based on traits of the organisms involved [3,4].

In particular, species' behavioural, morphological, physiological and life-history attributes influence the outcomes of environmental and biotic filters

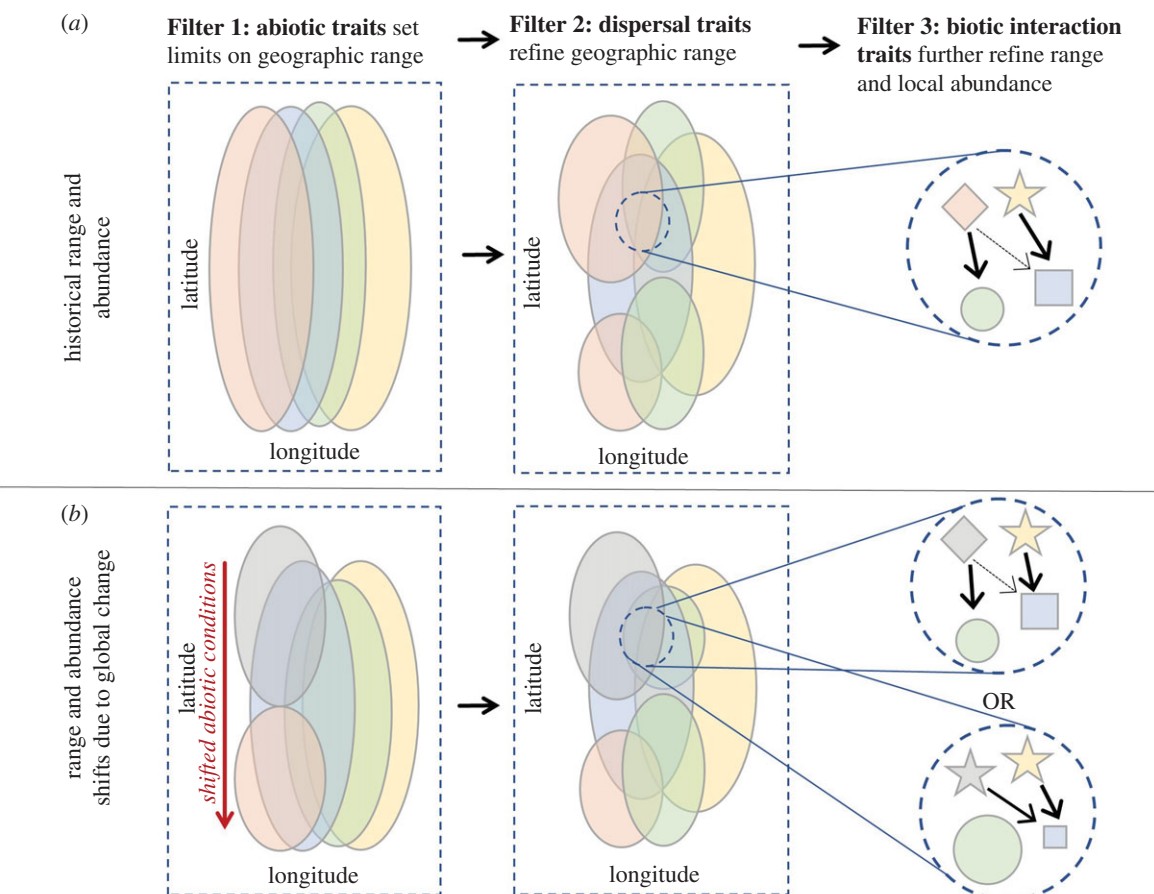

*(a)* **Filter 1: abiotic traits** set limits on geographic range → **Filter 2: dispersal traits** refine geographic range → **Filter 3: biotic interaction traits** further refine range and local abundance

*(b)*

*ellipse size/position = species geographic range; colour = species identity; shape = trait set; shape size = relative abundance; arrow thickness = strength of consumptive interaction within local food web.*

**Figure 1.** Trait-based filters of species range and abundance mediate the effects of global change on ecological communities. (*a*) Traits mediate species' interactions with their abiotic environment (e.g. thermal tolerance), setting limits on potential geographic ranges (i.e. fundamental niche space; Filter 1). Within this niche space, dispersal traits (e.g. larval duration) further restricts range (Filter 2). Finally, traits governing the strength and nature of biotic interactions (e.g. foraging mode) refine range and modify local abundance. (*b*) As abiotic conditions shift, *a prior* knowledge of species' traits and their influence on filters 1–3 allow us to predict changes in local species composition and abundance. For example, diamond versus star traits for the new grey species in *b*. (Online version in colour.)

on distribution and abundance across land and seascapes (figure 1). Species' characteristics that confer success under abiotic environmental conditions such as temperature, light, acidity, moisture (in terrestrial systems [5]) and dissolved oxygen (in aquatic systems [6]) provide the coarsest filter on geographic distribution (i.e. Filter 1 in figure 1; the boundaries of the fundamental niche). Traits that influence the outcome of biotic interactions within local environments such as organism size and shape [7] further mediate species persistence and co-existence (including trophic interactions; Filter 3 in figure 1). In addition, a host of traits confer information about species' dispersal ability [8] and mediate feedbacks between the effects of abiotic and biotic interactions on species' ranges and relative abundances (i.e. dispersal-limited controls on redistribution; Filter 2 in figure 1).

The need for predictive trait-based approaches is increasingly urgent due to mounting evidence that stresses like climate change, biological invasion and over-exploitation are profoundly affecting ecosystems and the socioeconomic benefits they provide. While ecological communities are inherently dynamic, with species membership and abundances varying over time and space [9,10], unprecedented anthropogenic stressors are driving species range and density changes exceeding historical levels [11]. Effects are proving to be unequal across species [12], so ecological communities are essentially being

pulled apart and reassembled with new member combinations [13]. Identifying species' characteristics that recur across unrelated taxa and confer information about species performance offers the potential to predict ecological dynamics as novel ecosystems form. For example, as abiotic conditions defining the fundamental niche shift under climate change, biotic traits that confer species' dispersal and survival abilities help define the shape of realized niches they are likely to occupy ([1,14,15]; figure 1*a* versus *b*).

Yet broad uptake of trait-based frameworks within global change ecology requires demonstrating they are an improvement over traditional species-based forecasts [16]—with forecasts extensively generated and compared via both methods. Given growing interest and research effort in the field of functional biogeography, we conducted a quantitative review of trait-based biodiversity research to evaluate the following three questions. (1) To what extent, and in what contexts, are trait-based insights being applied to predict the ecological outcomes of global change? (2) What research methods and frameworks show promise in moving the field from description and towards more nuanced prediction of community re-assembly and function? (3) What progress are we making in addressing barriers to predicting the outcome of environmental and biotic interactions under global change using functional traits?

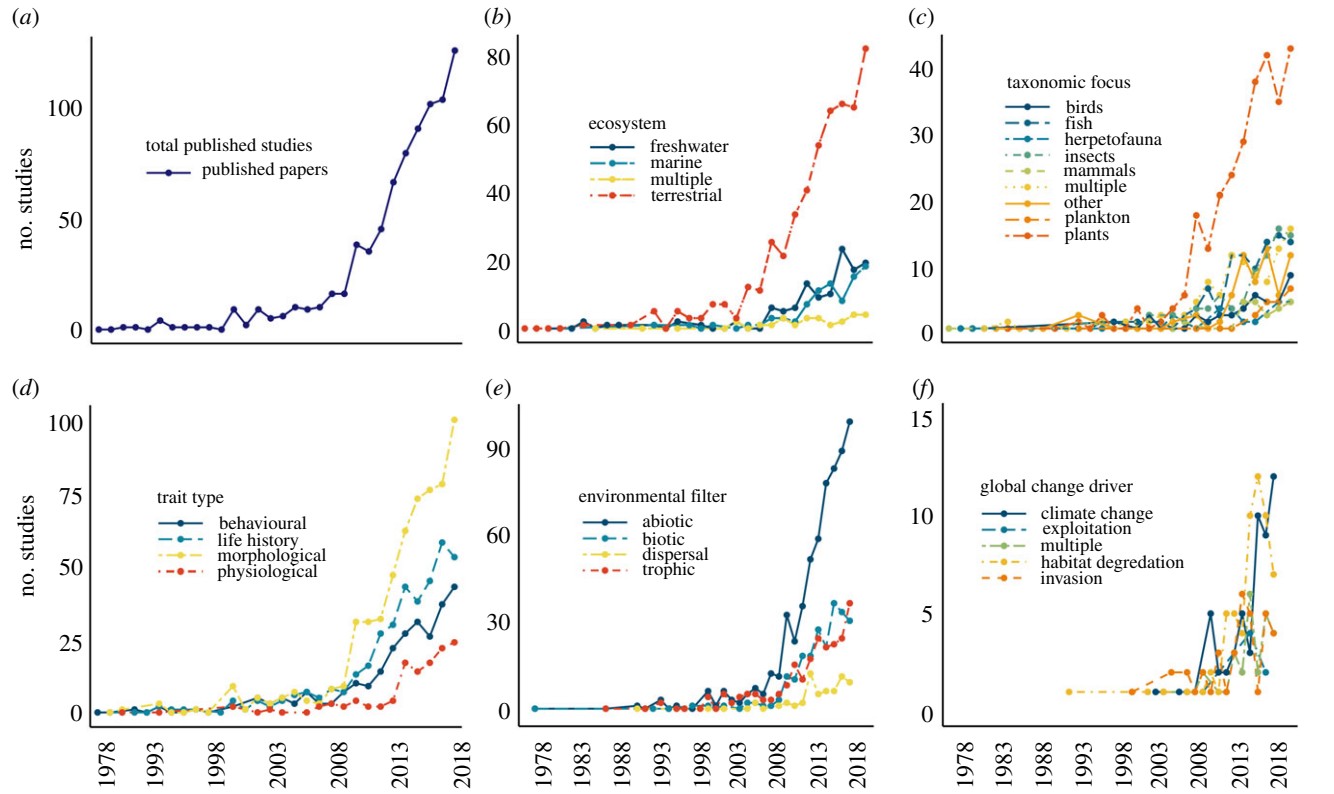

**Figure 2.** Temporal patterns of studies published in various categories over the years covered by our review. (Online version in colour.)

## 2. Reviewing trait-based global change ecology research

To synthesize current work on trait-based global change prediction, we conducted a systematic review of published literature via structured keyword searches in internationally recognized databases (electronic supplementary material S1, Literature search and classification procedures). We classified the results according to eight attributes: (1) environmental filter(s) on which the research focused (abiotic, dispersal or biotic interactions [further refined into trophic interactions]; figure 1); (2) type(s) of traits examined (life history, morphological, behavioural, physiological); (3) specific organismal traits quantified; (4) focal ecosystem; (5) taxonomic focus; (6) research methodology (experimental or observational study, reviews and meta-analyses or theoretical modelling); (7) whether a driver of global change was examined, and if so, type (e.g. climate change, biological invasion, habitat degradation); and (8) crucially, whether or not the study included predictions of ecological outcomes beyond the data set for which the analysis was initially constructed.

In addition to summarizing trends in trait-based research over time and in these eight domains, we visualized similarities and differences among the assemblages of traits across the studies through non-metric multidimensional scaling (nMDS) plots (electronic supplementary material S2, Multivariate analyses). Finally, we identified trait types contributing to differences between domains of trait-based research by constructing rank abundance curves and evaluating multivariate generalized linear models of the traits used in the literature, again grouped by factors 3–8 above (electronic supplementary material S2, Multivariate analysis).

## 3. Current trends in functional trait analyses of ecological phenomena

The volume of research applying traits to describe ecological phenomena has grown exponentially in the last decade (figure 2). In total, we identified 865 trait-based ecological studies published within greater than 200 journals as early as 1978, with more than half of this work coming from just 18 journals (electronic supplementary material, table S4). Exponential increases in trait-based studies since ∼2011 are dominated by observational studies relating variation in morphological and life-history features of vascular plants to abiotic filters of distribution and abundance within terrestrial ecosystems (figure 3; electronic supplementary material, figures S1, S11 and S13; 35% of all papers focused on plants, 30% on plant morphology). This trend is unsurprising given that trait-based research originated in plant community ecology (i.e. identifying morphological trait diversity and community-weighted mean values across abiotic gradients of light, moisture and nutrients [1]) but highlights a significant gap and opportunity to apply trait-based thinking to other systems and taxa. In contrast, trait-based investigations within marine and freshwater systems comprises just 30% of the studies combined, with a focus on observational studies of traits in the context of abiotic and dispersal filtering processes, primarily for planktonic taxa [17] and fishes [18] (figure 3; electronic supplementary material, figure S1).

Overall, trait-based research has overwhelmingly focused on organism size (figure 4; electronic supplementary material, table S3 and figures S4–S15). We identified 222 different size metrics used across studies—including vegetative height and cone length (plants), snout-ventral length or instar size

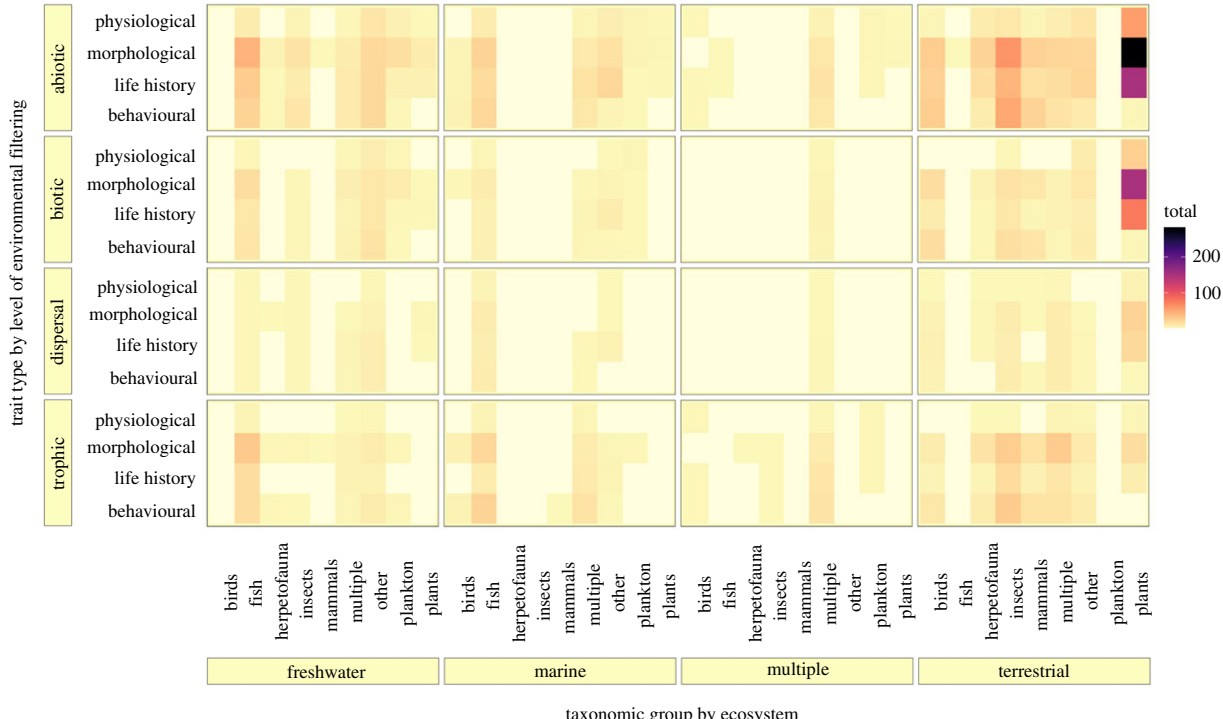

**Figure 3.** Trait-based ecological research effort by ecosystem type and taxonomic focus (*x*-axis), and level of environmental filtering and trait type (*y*-axis). Colour intensity relates to the number of research studies at the nexus of these domains, the smallest value on the plot is 1, where only one study is present for any given combination. (Online version in colour.)

(animals) and biovolume (cells) (electronic supplementary material, table S3). Body size is the most common trait measured in studies of animal taxa, and the most variable in terms of its definition and measurement, ranging from wet/dry mass to inference from organism length (electronic supplementary material, table S3). In contrast, SLA is the most common trait used within plant research and is estimated via a single standardized method across studies and plant taxa [5,19] (electronic supplementary material, table S3). In fact, SLA is the most common single trait examined in the studies reviewed here (135 occurrences over the 865 studies), with measures of body size a close second at 113 occurrences. While only 8% of traits (222) represented aspects of organism size, size-based traits were used within 57% of studies (495). The 222 size-based traits were used by 3.1 separate papers on average, while non-size traits were each used only 1.7 times on average.

Beyond size, a plethora of morphological ($n = 1{,}198$), behavioural ($n = 759$), life history ($n = 603$) and physiological (197 traits) features were applied to trait-based biodiversity research: in total, 2684 unique traits (figure 4; electronic supplementary material, tables S2 and S3). We attributed traits that confer information about the same process into conceptual groupings, revealing 203 'secondary' trait classifications (electronic supplementary material, table S3). Of these, 12% are only used in a single study while 32% are used in more than 10 studies (electronic supplementary material, table S3). Morphological traits include aspects of organisms' physical form (e.g. body shape, the presence and form of dentition or spines) and biochemical composition (e.g. nitrogen or carbon content; electronic supplementary material, table S3). Key behavioural traits include aspects of organisms' activity (e.g. movement rates or nocturnality) and habitat use (e.g. vertical habitat position within forest canopies or water columns, range size or edge position; electronic supplementary material,

table S3). Life history traits describe growth, abundance, survival and reproduction (including reproductive mode, timing and frequency), while physiological traits conferred information about organisms' environmental habitat requirements (e.g. moisture or temperature tolerances), and resource acquisition (e.g. photosynthetic rate; electronic supplementary material, table S3). Importantly, we observed significant variation in trait names (i.e. 'time to maturity' versus 'age at maturity'), and metrics of assessment (e.g. for trophic roles, guilds or positions) for the same property across studies (electronic supplementary material, table S3).

The diversity and identity of traits applied to research depend on the environmental filter under investigation (figure 1; figure 4*a–d*; electronic supplementary material, figures S8 and S9) and the ecosystem of interest (figure 4*i–k*; electronic supplementary material, figures S10 and S11). A far greater number of unique traits are used in plant compared with animal research (electronic supplementary material, figure S13; greater than 110 plant secondary trait classifications versus approx. 80 for fish and approx. 40 for mammals). Given the larger number of unique traits and greater abundance of studies focused on plants compared with animal taxa, one might expect the identity of traits to vary more among plant-focused studies compared with animal groups. Yet it appears the assemblage of traits is slightly more similar across plant papers compared with animals (i.e. slightly smaller convex hull in figure 5*a* and electronic supplementary material, figure S13 relative to both vertebrates and invertebrates; though all taxonomic groups contained the same general trait types but in slightly different proportions; electronic supplementary material, figures S2 and S12). We propose several potential explanations: (i) plant ecology has a longer history of using traits (figure 2), and thus overall may have made more progress distilling specific traits that represent key

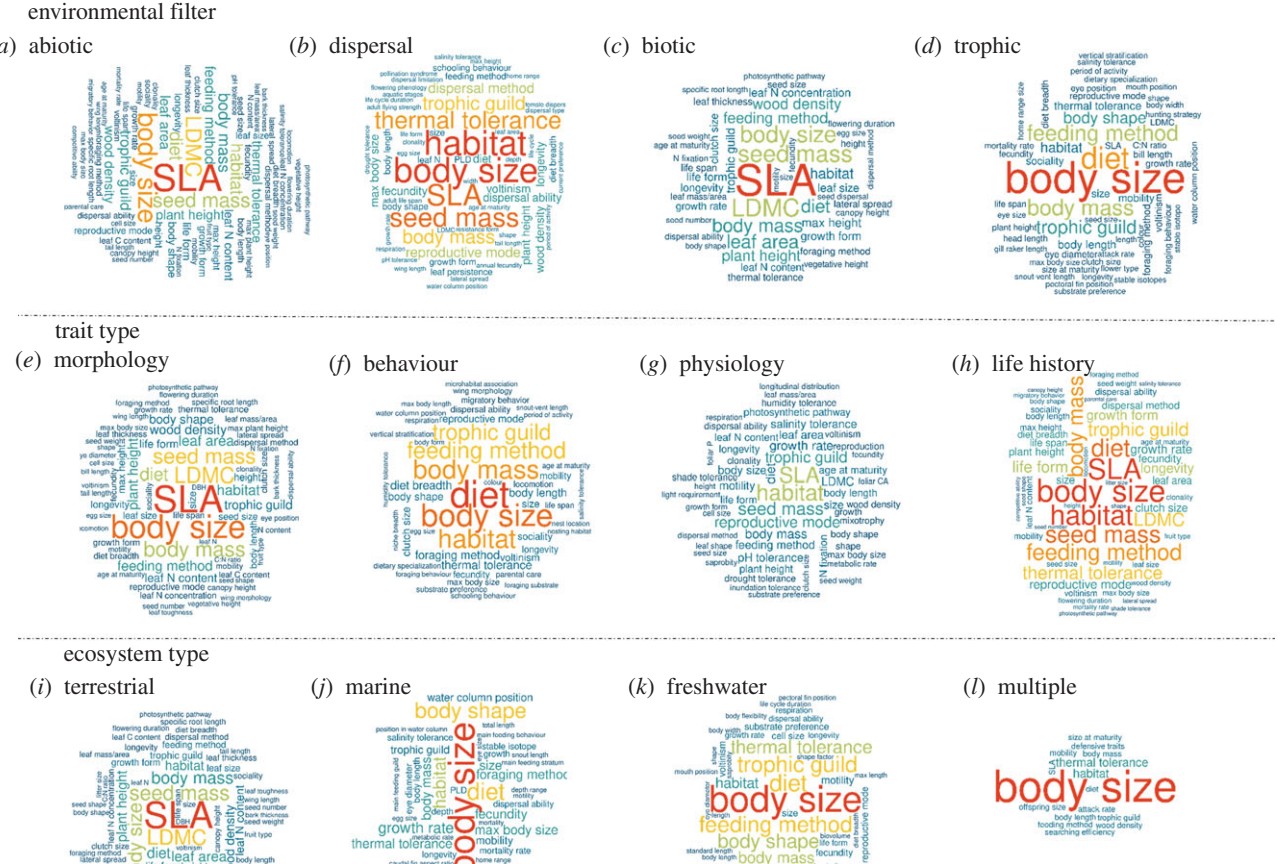

**Figure 4.** Top traits within ecological research on environmental filtering of species distribution and abundance. Word size is proportional to the relative number of studies focused on each trait by environmental filtering process. SLA = specific leaf area; PLD = pelagic larval duration; LDMC = leaf dry matter content. Multiple filters/trait types could be used by a single study. (Online version in colour.)

processes that are consistently used, (ii) traits that represent key processes have more standardized measurement methods than traits for other taxonomic groups (i.e. plant growth efficiency measured primarily as specific leaf area [SLA; leaf area per dry mass] versus the variety of animal body size measurements) and (iii) compared with mobile taxa, sessile plants and fungi (represented in 'other'; figure 5a, electronic supplementary material, figure S13) have a relatively narrower set of strategies for resource acquisition, defense, dispersal and reproduction.

Fewer traits have been applied in experiments, meta-analyses, reviews and theoretical work compared with observational research, for which we identified greater than 170 unique secondary trait classifications (electronic supplementary material, figure S15), and the identity of traits also varies among observational papers to a greater extent than other study types (i.e. greatest scatter in figure 5b). This is perhaps unsurprising given that observational work comprises most studies within the discipline and is likely starting place for first identifying and linking traits to important aspects of species distribution and interactions. Conversely, relatively few traits have been applied within multi-ecosystem studies (electronic supplementary material, figure S11), and the assemblage of traits is more similar among multi-ecosystem studies compared to ecosystem-specific research (i.e. smallest scatter in figure 5c). However, multi-ecosystem research primarily represents theoretical models (e.g. effect of food web structure and interaction strength on vulnerability to

extinction [20]), which necessitates selecting traits that can be estimated universally across taxa and systems such as body size, shape and feeding mode (figure 5c; electronic supplementary material, figure S14 and S15).

## 4. Current trends in global change prediction using functional traits

A small but increasing subset of studies occupy the nexus of predictive global change and trait-based ecology, and to date have primarily focused on traits related to abiotic habitat matching under climate change. Of the 865 functional trait studies, a small portion (23%) focused on applying traits in the context of global change (figure 2f), and even fewer (3.4%) applied traits to generate predictions beyond the data used for the initial analysis. Global change drivers, in order of decreasing frequency within the studies we evaluated, included habitat degradation (7.6%), climate change (6.6%), biological invasion (4.3%), multiple/non-specified drivers (3.4%) and exploitation (0.8%; figure 2f). Many of the global change studies take the classic approach of describing observed shifts in functional trait assemblages (e.g. shifts in Arctic fish traits with climate warming [21]).

Predictive studies emerged primarily within the last 10 years (79%); half of all predictive studies were published since 2015. More than a quarter (28%) of predictive studies focused on applying plant morphological traits to predict

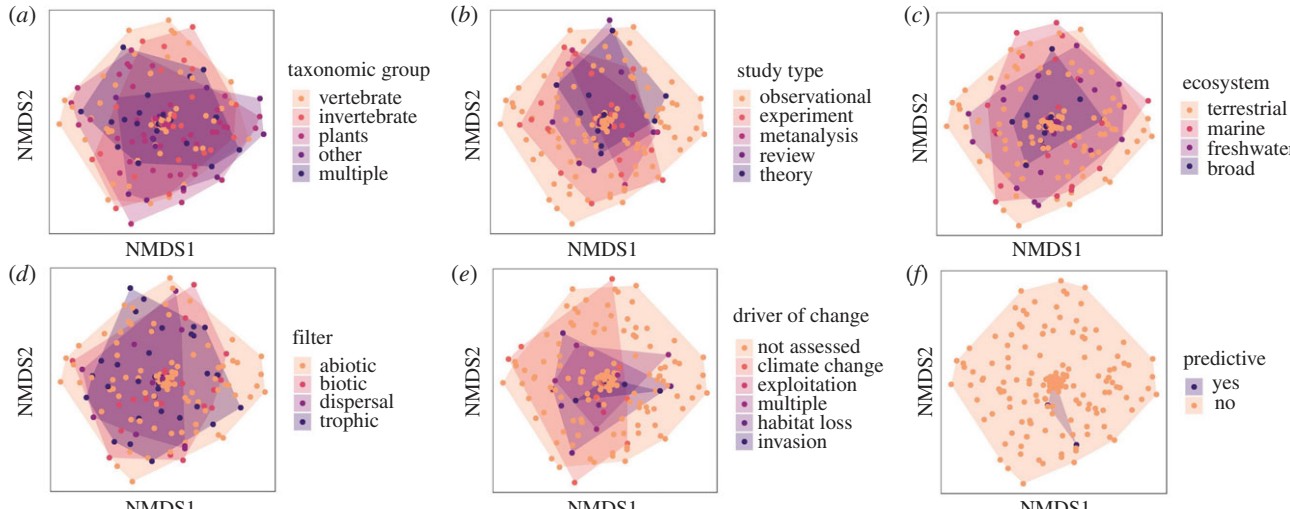

**Figure 5.** Non-metric multidimensional scaling (nMDS) plot visualizing the multivariate assemblage of traits applied within the ecological research we reviewed (each point = one study). Two-dimensional distance between points represents dissimilarity between multivariate trait sets in each study, calculated from a Bray-Curtis dissimilarity matrix of the presence or absence of each trait within each study. Shaded area represents the multivariate space occupied by studies in each level of the corresponding grouping factor. (Online version in colour.)

the outcomes of abiotic environmental filtering in terrestrial ecosystems (e.g. [22,23]). Crucially, studies that generated trait-based predictions of global change (the focus of this review) represent fewer than 3% (23) of all studies (electronic supplementary material, table S5). Of these, more than half (13) focus on ecological prediction in a climate change context, three on biological invasions, and a single study each on the consequences of habitat degradation and exploitation. Five studies used traits to predict the outcomes of multiple global change drivers (two marine [24,25]; three terrestrial [23,26,27]).

Traits applied within predictive global change studies are highly nested within the broader suite of traits used across non-global change studies (figures 5e,f and 6; electronic supplementary material, figures S2E/F and figure S3E/F). Again, organism size was the most common single trait. Habitat associations and life history were the most important suites of traits in studies investigating climate change effects, while morphological traits were most important to studies of habitat degradation and biological invasions (figure 6; electronic supplementary material, figure S4 and S5, table S5, tables S6–S9). In general, physiological traits related to resource acquisition and requirements, such as thermal tolerance, and life-history traits are more often applied within predictive studies compared with descriptive work (figure 6; electronic supplementary material, figures S6 and S7, table S5, S10–13).

## 5. Promising methodological frameworks for trait-based global change prediction

Trait-based methods vary in scale and the complexity of required data inputs, and therefore result in outputs at a range of resolutions (figure 6). The predictive global change studies we identified allow us to assess a range of trait-based methodologies (experiments, statistical models, meta-analyses and process models [spatial and non-spatial]) in terms of their likelihood of generating output that informs biodiversity conservation and management decision-making (figure 6; electronic supplementary material, table S5). We suggest that

researchers select among frameworks for predicting ecological outcomes under future environmental conditions based on whether the method: (1) explicitly integrates mechanistic knowledge or hypotheses about the environmental filter(s) likely to be disrupted by focal global change driver[s] (figure 1), (2) considers scales (taxonomic, spatial, and temporal) at which functional trait and environmental condition information are required and available for the focal system, and (3) whether the forecasting output (e.g. point estimate or probability distribution, threshold value or limit, spatially or temporally explicit) is relevant for the specific conservation or management interventions being considered.

Experiments and statistical models of observational data offer opportunities to generate and test fine-scale predictions about response to global change drivers (e.g. figure 6; eelgrass communities under climate change and grazer loss [25]). However, insights gained through experimentation are most relevant under the set of conditions under which the study takes place. Likewise, predictions from statistical models of trait-environment relationships (the most common type; figure 6) are bounded by the conditions under which observations are made, and thus of limited utility in cases where systems are pushed outside their range of historical variation. However, as bodies of experimental and observational work grow, results can be synthesized via meta-analyses to generalize relationships between global change drivers and effect sizes (e.g. in terms of change in abundance or distribution) for trait types that recur across taxa and ecosystem type (e.g. synthesis of acidification effects on marine fishes [28]; figure 6).

Trait-based species distribution models are spatially explicit methods for generating ecological predictions of future species' range and abundances under future environmental conditions (i.e. abiotic filtering; figure 1, Filters 1 and 2). To date, the majority of work combining this method with traits frameworks has focused on distribution under future climate conditions (e.g. figure 6; distribution models of trees [29] and freshwater fishes [30]). Spatial projections generated from distribution models can be intuitively applied to place-based biodiversity conservation and natural resource management, but generally omit biotic interactions and feedbacks that

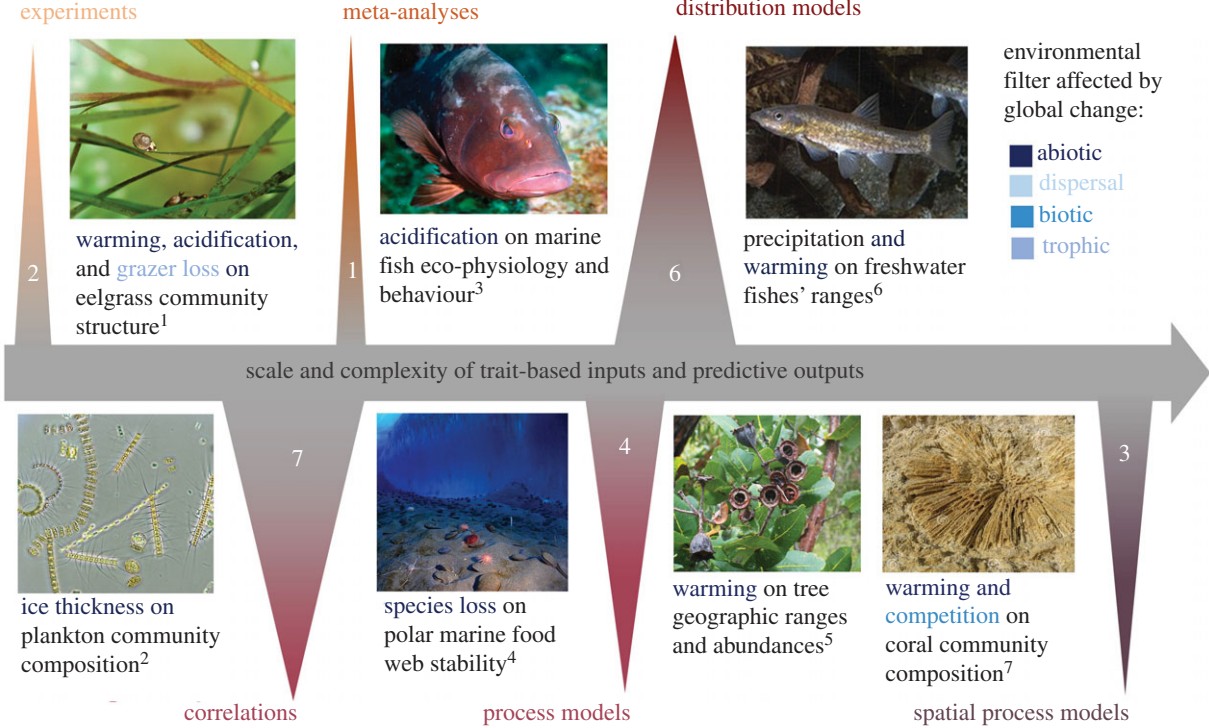

**Figure 6.** Subset of studies explicitly predicting future ecological outcomes under global change based on trait-based interactions and the types of environmental filtering they consider. Approaches vary in scale and resolution of trait data inputs and predictive outputs. Numbers inset in triangles and relative triangle size represents the number of studies in each type among 23 predictive global change papers; trait-based correlation and distribution models are most common. Though likely to generate the most robust predictions in forms useful for conservation planning, spatially explicit trait-based process models are relatively uncommon. Superscript numbers denote, [1][25], [2][17], [3][28], [4][24], [5][29], [6][30], [7][31]. (Online version in colour.)

further refine species' ranges and abundances across the landscape (figure 1, Filter 3).

Trait-based process models offer a means to incorporate biotic interactions into forecasts of future ecological states. Functional biogeographers have consistently recognized the need to move away from modelling pattern and towards process [32]; for example, away from reliance on tracking the frequency and mean value of traits that might be a proxy for the outcomes of competition within a community (e.g. tree height [1]) and towards modelling interactions directly based on traits known to affect competitive ability (e.g. nutrient tolerance, photosynthetic rates, root morphology). While process models offer an opportunity to examine the effect of more complex interactions on ecological phenomena under global change, such models are often not spatially- or temporally-explicit. As a result, process models often produce output that is mismatched with the scales and units required for conservation and management decision-making (e.g. figure 6; trait-based polar sea food web model [24]).

Ultimately, spatially- and temporally-explicit process models are likely to be the most promising techniques for global change prediction because they offer a means to generate range and abundance projections that account for multiple environmental filtering processes simultaneously, including biotic interactions (i.e. figure 1, Filters 1–3), and produce outputs that can be adapted to the resolution required for decision-making. Such models are increasingly used to forecast ecosystem states under alternative assumptions about global change (e.g. simultaneous effects of harvest and climate on food webs and the fisheries they support [33]). Morphological traits such as organism size and physiological traits such as thermal performance has been broadly applied within spatially-explicit process models (e.g. figure 6; coral survival

and growth under climate states in marine systems [31] and vegetation models in terrestrial systems [23]). Our synthesis highlights a range of traits known to influence environmental filtering (figure 4; electronic supplementary material, tables S2 and S3) that could also be used to forecast the strength of biotic interactions within spatial process models. Below we discuss research directions that will bolster the development of trait-based, spatially explicit process models for global change prediction.

# 6. Persistent challenges and opportunities for trait-based global change prediction

Despite growing interest in trait-based frameworks for describing ecological patterns (figure 2), our review highlights a significant opportunity to bolster their application to global change prediction (figure 5*f*). Realizing the full potential of frameworks like spatially-explicit trait-based process models requires continued progress on at least three challenges consistently highlighted in the burgeoning field of functional biogeography [1,3,4]: (1) increasing the use of multivariate (rather than univariate) trait assemblages to describe ecological processes, (2) consistently matching scales of environmental data collection to trait-based process(es); (3) propagating intraspecific trait and environmental variation into forecasts.

## (a) Utilizing multivariate trait 'syndromes' for global change prediction

Organisms' responses to their environment are governed by complex suites of correlated traits that confer information

about performance under specific sets of environmental filters (figure 1). Strong correlation among trait types that can recur across unrelated species—trait 'syndromes' or 'typologies'—underpin trait-based community assembly theory for plants [3] and behavioural syndromes in animals [34]. Yet single trait-type studies make up roughly half of the research we reviewed (412 papers) compared with multi-trait (i.e. three or more traits) studies (182 papers, 21%). In considering a single trait at a time as a function of a species' or ecological community's response to gradients of change, we risk overlooking the combined effect that a range of traits may have in explaining responses to global change drivers (e.g. multivariate traits for restoration design to resist invasion [35], and plant trait typologies along ecological gradients [4], or emergent functions [36]).

In particular, body size scales with key processes across all stages of environmental filtering ([7]; e.g. desiccation tolerance, metabolism, prey consumption rates, movement), has been of great benefit for describing ecological structure and function [37] and needs to be accounted for in analyses using autocorrelated traits. However, a substantial amount of variation in key processes is not explained by size [38]. Moreover, other morphological and physiological traits mediate the influence of size, such as shape and metabolic approach to temperature regulation. Within size classes, variation in key behavioural and life-history traits influence establishment and persistence within ecological communities; for example, reproductive behaviours are often incorporated into trait-based analyses that model species' and populations' dispersal capabilities in changing and novel ecosystems [39]. Greater inclusion and uptake of traits in analyses other than size—for instance metabolic, fecundity, ontological, growth traits—can increase our ability to predict future ecological states arising from ongoing global change.

Increasingly sophisticated statistical tools and computing power enable greater complex multi-trait analyses of ecological relationships [16]; for example, multi-matrix modelling to simultaneously assess relationships between species abundances and/or distributions, environmental gradients, and key traits [40,41]. While continuous trait variables are more tractable for modeling, classifying taxa into discrete (i.e. categorical) trait groups may offer convenience, especially for conservation prioritization [42,43]. The list of traits synthesized in this review could provide a starting place for researchers who seek to identify suites of organism attributes related to ecological processes within their research (electronic supplementary material, tables S2 and S3).

## (b) Matching the scale of environmental data collection to trait-driven process(es)

Species traits are often measured along major environmental gradients (e.g. temperature, acidity, soil quality, wind, etc. along elevation or latitude), with measurements focused on capturing intraspecific variation [44], interspecific variation [45], or both [46]. Designing field data collection with trait and environmental data sets gathered at equal resolutions allows ecologists to quantify the scale at which variation in both response and explanatory variables matters for the environmental filtering process(es) under investigation (i.e. figure 1). Yet trait variation is often not explicitly linked to measures of the important aspect of the gradient at the same resolution (e.g. environmental data collected at the region or site level, while trait data collected at the individual

level [16] or vice versa). Moreover, many of the observational studies we reviewed lacked explicit hypotheses about the mechanism(s) linking variation in species' abundances and distributions to the traits under investigation, which are necessary for selecting the relevant scale(s) for data collection. In such cases, advanced multivariate techniques (e.g. RLQL analysis, the fourth corner solution [41] and hierarchical modelling of species communities [HMSC]) can be used to parse out the spatial and temporal scale at which relationships between traits and environmental gradients hold, helping researchers identify potential ecological mechanisms driving trait patterns across land and seascapes [47].

## (c) Accounting for intraspecific trait variation in predicted responses to global change

Few studies (38 papers; 4%) specifically investigate intraspecific trait variation and none of the predictive global change studies we identified. In practice, the acquisition of high-resolution trait information measured for individuals within populations is labour-intensive and specific to a temporal, spatial and ecological context (e.g. lipid content or energy density of prey species). Aggregate values, when available, are often used at the population or species resolution to model broader patterns in the responses to environmental or ecological variables (e.g. [43]). However, individuals within a population often possess traits that confer advantages for dispersal or persistence within changing ecosystems [48] and can vary across ontogeny. Trait-based modelling may therefore overlook nuanced ecological processes when intraspecific variation is ignored; highlighting a trade-off between trait relevance and data collection effort.

Our synthesis highlights the need for continued international efforts to aggregate and make accessible trait information, and in particular intraspecific variation and associated environmental covariates at local and regional scales. Prominent trait databases include FishBase and SeaLifeBase [49] for fishes and marine invertebrates, respectively, and TRY [50] for plants. But many studies report undertaking significant additional manual curation of the information acquired from these databases or require trait resolution at a finer scale than is available—efforts that are labour-intensive, costly, and often require consultation with taxonomic experts, and thus represents a significant barrier to expanding trait-based approaches. Online repositories designed to facilitate the collation of fine-scale, within-population trait data from local and regional collection efforts will help to address these barriers. Such efforts require sustained baseline funding to maintain repositories and evolve products to address emerging needs [51].

## 7. Re-examining the past to inform future trait-based predictions

The Anthropocene is characterized by unprecedented changes to ecosystems. There is an urgent need to synthesize current trajectories of change, predict future ecological outcomes in relation to multiple drivers of change, and account for the naturally large number of components affected by and effecting change. Trait-based approaches can help reduce complexity and offer relief to many technical challenges in these pursuits [16]. The persistence or loss of

species in novel ecosystems will depend on several factors that may be predicted using traits: (1) species' potential responses to environmental forcing (traits describing dispersal, establishment, persistence), (2) the capacity of species to affect community dynamics (i.e. interaction strengths) and (3) the combined effect of multiple anthropogenic forces on organisms' interactions with the environment and one another (i.e. either additive, antagonistic, synergistic) and (4) the type and duration of stressors (e.g. [52]). Many more applications of trait-based forecasting to assess ecological change are needed, as well as validations of such models compared to taxonomic approaches. Ecologists are well-positioned to build and test trait-based predictions through hindcasting ecological outcomes in systems that continue to face rapid community reassembly; For example, climatic transition zones—regions at the boundaries between tropical and temperate ecosystem types in Australia, Japan, the Eastern Pacific and Western Atlantic [53]—are areas where suites of species are readily being redistributed due to environmental forcing. These ecological mixing zones provide excellent opportunities to test trait-based hypotheses of rapid ecological change across gradients in natural experimental settings and where traits have explained shifts in the distributions and abundances of range expanders (electronic supplementary material, S3: Case Study; electronic supplementary material figure S16).

Data accessibility. Project data and code are publicly accessible on GitHub at https://github.com/CHANGE-Lab/traits-review.
The data are provided in electronic supplementary material [54].

Authors' contributions. S.J.G.: conceptualization, data curation, formal analysis, funding acquisition, methodology, project administration, supervision, visualization, writing—original draft, writing—review and editing; C.B.B.: data curation, formal analysis, methodology, visualization, writing—original draft, writing—review and editing; N.A.H.: conceptualization, formal analysis, methodology, visualization, writing—original draft, writing—review and editing; L.B.C.: conceptualization, funding acquisition, writing—review and editing.

All authors gave final approval for publication and agreed to be held accountable for the work performed therein.

Competing interests. We declare we have no competing interests.

Funding. This research was supported by Lenfest Ocean Program Grant to S.J.G. and L.B.C. and a Sloan Research Fellowship to S.J.G.

Acknowledgements. We are grateful to participants in the 2019 Traitspace workshop in for feedback and discussion that has greatly enriched and improved this study. Figure images credited to M. Bell, S. Anderson, H. Outlaw, M. Donaldson, B. Gratwick and NOAA Ocean Exploration & Research.

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
