## [Peer Review File · Proceedings of the Royal Society B: Biological Sciences]

Review History

RSPB-2021-1277.R0 (Original submission)

Review form: Reviewer 1

Recommendation

Major revision is needed (please make suggestions in comments)

Scientific importance: Is the manuscript an original and important contribution to its field?

Good

General interest: Is the paper of sufficient general interest?

Excellent

Quality of the paper: Is the overall quality of the paper suitable?

Good

Is the length of the paper justified?

Yes

Should the paper be seen by a specialist statistical reviewer?

No

Do you have any concerns about statistical analyses in this paper? If so, please specify them explicitly in your report.

No

It is a condition of publication that authors make their supporting data, code and materials available - either as supplementary material or hosted in an external repository. Please rate, if applicable, the supporting data on the following criteria.

Is it accessible?

Yes

Is it clear?

N/A

Is it adequate?

N/A

Do you have any ethical concerns with this paper?

No

Comments to the Author

Review of „Trait-based approaches to global change ecology: moving from description to prediction“ by Green et al.

This study is a literature study in the field of trait-based ecology. The authors focussed on studies dealing with global change and they were particularly interested in studies making predictions in the light of global change. This is a very timely and important task. Many trait-based studies have been published in the last decades and the time is right to get an overview and to guide research to the most urgently needed questions. A total number of 865 studies was analyzed and a huge amount of data on traits, taxa, habitats etc. was compiled. One result was that in only very few studies trait-based approaches were used to make predictions in response to global change. I very much appreciate the approach, but I see some room for improvements.

- a) In my opinion, the reader would benefit from more examples of individual studies and the outcome of those studies. For example (not restricted to that) in the part from line 150 to 155. The few predictive studies are very well presented in combination with Figure 6.
- b) I was thinking if there is a causal relationship between the majority of size as a trait (and specific leaf area) used and the low number of predictive studies. Size is relatively easy to measure, it is an important „master trait“, however, a change in size itself does not tell necessarily a lot about a change in ecosystem function; in particular if this change is small. For predictions, other and multiple traits than size might be needed.
- c) What was the focus of the many studies that were not making predictions? Did they simply fail or did they have a different focus? In phytoplankton, for example, it was found that in marine studies, trait-based approaches are often used to explain biogeochemical processes, whereas in freshwater studies, trait-based approaches are used to understand the community response to environmental factors/changes (Weithoff & Beisner, 2019, Front Mar Sci). Could you see similar distinctions in your data set? Would be interesting.
- d) The meaning of narrow assemblages needs more explanation. What does it tell the reader, when the trait assemblages are narrow? And what is the ecological meaning? Which traits were used in the studies that are referred to in line 99-107?
- e) The term „novel ecosystems“ appears in the abstract and the conclusive part 7 and only once in the text. Do the authors regard any ecosystem affected by global change as a novel ecosystem? I think the term „novel ecosystems“ needs a definition when used, because for different readers, it might have different meanings. Maybe use Hobbs et al. 2009, TREE as a reference.
- f) „Plankton“ is not a taxon. Plankton can be plants (phytoplankton) or animals (zooplankton) or something in between (mixotrophs), or procaryotes. I think it needs to be

justified, why no taxonomic distinction was made.

g) Line 76 I couldn't find a table displaying the journals where the studies were published in.

h) Figure 3. Can you change the colour code so that 0 (zero) is white. At first glance it looks like quite a number of studies for all combinations.

Review form: Reviewer 2

Recommendation

Reject – article is not of sufficient interest (we will consider a transfer to another journal)

Scientific importance: Is the manuscript an original and important contribution to its field?

Acceptable

General interest: Is the paper of sufficient general interest?

Good

Quality of the paper: Is the overall quality of the paper suitable?

Acceptable

Is the length of the paper justified?

Yes

Should the paper be seen by a specialist statistical reviewer?

No

Do you have any concerns about statistical analyses in this paper? If so, please specify them explicitly in your report.

No

It is a condition of publication that authors make their supporting data, code and materials available - either as supplementary material or hosted in an external repository. Please rate, if applicable, the supporting data on the following criteria.

Is it accessible?

Yes

Is it clear?

No

Is it adequate?

Yes

Do you have any ethical concerns with this paper?

No

Comments to the Author

Review of RSPB-2021-1277

This manuscript reviewed literature on studies of global change effects on organisms utilising (functional) traits. It's scope is broad, ranging across biomes and taxa as broadly as possible. The manuscript's aims are most clearly articulated in the last paragraph of Section 1. The manuscript uses a systematic review to query the literature and provide an empirical basis for questions of how are traits being used, which traits, across different fields and with respect to various

processes. The review also aims to highlight beneficial ways forward to provide a predictive basis to support management decision regarding ecosystem function and community assembly in response to global change drivers.

The first question is straightforward. What does the bibliometric data tell us? The authors have comprehensively and clearly communicated the results of this enquiry. The next two aims I found harder to parse and to judge whether the authors had achieved them. “(2) What progress are we making in addressing potential barriers to predicting the outcome of environmental and biotic interactions using functional traits?

It is a little vague. And how is progress measured? a judgement? I could not assess whether this had been done well.

And the 3rd, “What research methods and frameworks show promise in moving the field from description and towards more nuanced prediction of community re-assembly and function?: How is ‘promise’ judged?

But those quibbles with wording of aims aside, I am not clear how to judge this work. While the manuscript’s breadth is a strength, it perhaps is the study’s weakness, because it leads to some broad statements that come across as quite vague. On the one hand I am sympathetic to the aim of using traits in models, to be predictive. And I see the merit of reviewing the state of play. And yet I feel that probably considerable predictive modelling is missed by this review, because of the breadth addressed. It also has resulted in a lack of specificity, in my opinion.

It is my belief that traits are useful in a comparative context to assess the capacity for, and to facilitate, generalisation. This aspect appears to be ignored or at least not highlighted. Instead, the highlight is placed on using traits in (modelling effects of) global change. Now, I agree with that research program of using traits to predict processes and outcomes. And I agree entirely with the need to match scales of investigation (drivers and responses) and the outcomes of importance to managers. However, broad-scale exploratory studies (correlations) enable the basis for fine-scale investigations and predictions to be made. To dismiss correlative descriptive studies of traits seems to me to miss the point.

By contrast, in at least one of the papers highlighted as a good example of using traits to predict change, no mention is made of where those parameters came from (ref 28. Fordham et al 2012). Often demographic models are parameterised with best guesses and observations (of demographic rates), the promise of traits would be to provide estimates of those rates from measurable traits (e.g. extending the exploratory study of Adler et al 2014 <https://doi.org/10.1073/pnas.1315179111>).

While specifying traits, and then accepting a very wide range of traits (eg habitat type, competitive ability (specified how?)), the authors appeared to not include functional groups. Which is curious because much (predictive) global change modelling (e.g. dynamic global vegetation dynamics modelling) has used functional type approaches. So I would expect some discussion of how functional groups or types have been used in modelling and why traits should be considered different or better from them and with specific reference to the purpose and constraints of predictive modelling. My own feeling (only opinion) is that specifying types offers convenience for modelling and its communication, relative to continuous traits.

I found it interesting that a recent review (2019) of trait-based modelling was not mentioned (Zhakarova <https://doi.org/10.1016/j.ecolmodel.2019.05.008>) even though the topics overlap considerably. In my opinion that paper, focussed on modelling rather than global change, is more explicit about what has been done and why and how to proceed.

Specific minor comments

Abstract What does it mean to “propagate environmental (and trait) variation throughout process model functions”?

L57&ff how did you deal with line cases? morphological vs physiological?

L76 this does not match the Supplement Table S4

L93 SLA is not a measure of plant size. it is a measure of the leaf area deployed for carbon gain, per unit of mass invested. A very different measure. It could be thought of as an 'efficiency'. Plant size is often measured by height or lateral spread or stem diameter or biomass. Leaf Area might be thought of a measure of size, within a restricted domain of plants, especially herbs and grasses, and mostly within species rather than between species.

LL150 & ff this is the nub. Very few trait-based predictive models in response to global change drivers. Though this might neglect older functional type research, especially in the vegetation dynamics field.

LL162-164 and are they associated with fewer species?

I found Fig 6 a bit empty. It really just seemed a collection of pictures and some aspects of studies. 'spatially-explicit', experimental, correlations. I don't find the use of triangles as indicative of number of studies very clear or compelling.

Sect 6 LL241& ff. ok, but isn't that saying just do more of everything. Perhaps there are trade-offs in what approaches are taken. And just including more information, does not guarantee a better outcome.

LL295 & ff. (also LL272-277) I don't get the point here. Yes, there are statistical methodologies to relate traits, environment and performance from observational data. What is the point of the last sentence? are you saying such methods don't work? Are not enough? Or something else, in which case please be explicit about it. Are you proposing path analysis, for instance.

LL349 & ff this section would seem to benefit from some references.

Decision letter (RSPB-2021-1277.R0)

13-Jul-2021

Dear Dr Green:

I am writing to inform you that your manuscript RSPB-2021-1277 entitled "Trait-based approaches to global change ecology: moving from description to prediction" has, in its current form, been rejected for publication in Proceedings B.

This action has been taken on the advice of referees, who have recommended that substantial revisions are necessary. The referees do, however, differ in their view of whether the revisions recommended would take the manuscript above the acceptance threshold, with referee 1 being more positive than referee 2. While some of referee 2's negativity comes from (what the referee freely admits are) personal opinions rather than 'errors' in the manuscript, if you want your paper to have broad impact, as you surely do, then this is the sort of person you need to win over. For example, I think you should read the review by Zakharova et al. (2019) because any assessment of the value of trait-based approaches for global change ecology will gain from understanding the value of trait-based approaches in ecology per se. If you think you can make changes that would fully address the comments of both referees, I would be happy to consider a resubmission. However please note that this is not a provisional acceptance.

The resubmission will be treated as a new manuscript. However, we will approach the same reviewers if they are available and it is deemed necessary to do so. Please note that resubmissions must be submitted within six months of the date of this email. In exceptional circumstances,

extensions may be possible if agreed with the Editorial Office. Manuscripts submitted after this date will be automatically rejected.

- 1) A 'response to referees' document including details of how you have responded to the comments, and the adjustments you have made.
- 2) A clean copy of the manuscript and one with 'tracked changes' indicating your 'response to referees' comments document.
- 3) Line numbers in your main document.
- 4) Please read our data sharing policies to ensure that you meet our requirements <https://royalsociety.org/journals/authors/author-guidelines/#data>.

Best wishes,
Innes Cuthill

Prof. Innes Cuthill
Reviews Editor, Proceedings B
mailto: proceedingsb@royalsociety.org

Reviewer(s)' Comments to Author:

Referee: 1

Comments to the Author(s)

Review of „Trait-based approaches to global change ecology: moving from description to prediction“ by Green et al.

This study is a literature study in the field of trait-based ecology. The authors focussed on studies dealing with global change and they were particularly interested in studies making predictions in the light of global change. This is a very timely and important task. Many trait-based studies have been published in the last decades and the time is right to get an overview and to guide research to the most urgently needed questions. A total number of 865 studies was analyzed and a huge amount of data on traits, taxa, habitats etc. was compiled. One result was that in only very few studies trait-based approaches were used to make predictions in response to global change. I very much appreciate the approach, but I see some room for improvements.

- a) In my opinion, the reader would benefit from more examples of individual studies and the outcome of those studies. For example (not restricted to that) in the part from line 150 to 155. The few predictive studies are very well presented in combination with Figure 6.
- b) I was thinking if there is a causal relationship between the majority of size as a trait (and specific leaf area) used and the low number of predictive studies. Size is relatively easy to measure, it is an important „master trait“, however, a change in size itself does not tell necessarily a lot about a change in ecosystem function; in particular if this change is small. For predictions, other and multiple traits than size might be needed.
- c) What was the focus of the many studies that were not making predictions? Did they simply fail or did they have a different focus? In phytoplankton, for example, it was found that in marine studies, trait-based approaches are often used to explain biogeochemical processes, whereas in freshwater studies, trait-based approaches are used to understand the community response to environmental factors/changes (Weithoff & Beisner, 2019, Front Mar Sci). Could you see similar distinctions in your data set? Would be interesting.

- d) The meaning of narrow assemblages needs more explanation. What does it tell the reader, when the trait assemblages are narrow? And what is the ecological meaning? Which traits were used in the studies that are referred to in line 99-107?
- e) The term „novel ecosystems“ appears in the abstract and the conclusive part 7 and only once in the text. Do the authors regard any ecosystem affected by global change as a novel ecosystem? I think the term „novel ecosystems“ needs a definition when used, because for different readers, it might have different meanings. Maybe use Hobbs et al. 2009, TREE as a reference.
- f) „Plankton“ is not a taxon. Plankton can be plants (phytoplankton) or animals (zooplankton) or something in between (mixotrophs), or procaryotes. I think it needs to be justified, why no taxonomic distinction was made.
- g) Line 76 I couldn't find a table displaying the journals where the studies were published in.
- h) Figure 3. Can you change the colour code so that 0 (zero) is white. At first glance it looks like quite a number of studies for all combinations.

Referee: 2

Comments to the Author(s)

Review of RSPB-2021-1277

This manuscript reviewed literature on studies of global change effects on organisms utilising (functional) traits. It's scope is broad, ranging across biomes and taxa as broadly as possible. The manuscript's aims are most clearly articulated in the last paragraph of Section 1. The manuscript uses a systematic review to query the literature and provide an empirical basis for questions of how are traits being used, which traits, across different fields and with respect to various processes. The review also aims to highlight beneficial ways forward to provide a predictive basis to support management decision regarding ecosystem function and community assembly in response to global change drivers.

The first question is straightforward. What does the bibliometric data tell us? The authors have comprehensively and clearly communicated the results of this enquiry. The next two aims I found harder to parse and to judge whether the authors had achieved them. “(2) What progress are we making in addressing potential barriers to predicting the outcome of environmental and biotic interactions using functional traits?

It is a little vague. And how is progress measured? a judgement? I could not assess whether this had been done well.

And the 3rd, “What research methods and frameworks show promise in moving the field from description and towards more nuanced prediction of community re-assembly and function?: How is ‘promise’ judged?

But those quibbles with wording of aims aside, I am not clear how to judge this work. While the manuscript's breadth is a strength, it perhaps is the study's weakness, because it leads to some broad statements that come across as quite vague. On the one hand I am sympathetic to the aim of using traits in models, to be predictive. And I see the merit of reviewing the state of play. And yet I feel that probably considerable predictive modelling is missed by this review, because of the breadth addressed. It also has resulted in a lack of specificity, in my opinion.

It is my belief that traits are useful in a comparative context to assess the capacity for, and to facilitate, generalisation. This aspect appears to be ignored or at least not highlighted. Instead, the highlight is placed on using traits in (modelling effects of) global change. Now, I agree with that research program of using traits to predict processes and outcomes. And I agree entirely with the need to match scales of investigation (drivers and responses) and the outcomes of importance to managers. However, broad-scale exploratory studies (correlations) enable the basis for fine-scale investigations and predictions to be made. To dismiss correlative descriptive studies of traits seems to me to miss the point.

By contrast, in at least one of the papers highlighted as a good example of using traits to predict change, no mention is made of where those parameters came from (ref 28. Fordham et al 2012). Often demographic models are parameterised with best guesses and observations (of demographic rates), the promise of traits would be to provide estimates of those rates from measurable traits (e.g. extending the exploratory study of Adler et al 2014 <https://doi.org/10.1073/pnas.1315179111>).

While specifying traits, and then accepting a very wide range of traits (eg habitat type, competitive ability (specified how?)), the authors appeared to not include functional groups. Which is curious because much (predictive) global change modelling (e.g. dynamic global vegetation dynamics modelling) has used functional type approaches. So I would expect some discussion of how functional groups or types have been used in modelling and why traits should be considered different or better from them and with specific reference to the purpose and constraints of predictive modelling. My own feeling (only opinion) is that specifying types offers convenience for modelling and its communication, relative to continuous traits.

I found it interesting that a recent review (2019) of trait-based modelling was not mentioned (Zhakarova <https://doi.org/10.1016/j.ecolmodel.2019.05.008>) even though the topics overlap considerably. In my opinion that paper, focussed on modelling rather than global change, is more explicit about what has been done and why and how to proceed.

Specific minor comments

Abstract What does it mean to “propagate environmental (and trait) variation throughout process model functions”?

L57&ff how did you deal with line cases? morphological vs physiological?

L76 this does not match the Supplement Table S4

L93 SLA is not a measure of plant size. it is a measure of the leaf area deployed for carbon gain, per unit of mass invested. A very different measure. It could be thought of as an ‘efficiency’. Plant size is often measured by height or lateral spread or stem diameter or biomass. Leaf Area might be thought of a measure of size, within a restricted domain of plants, especially herbs and grasses, and mostly within species rather than between species.

LL150 & ff this is the nub. Very few trait-based predictive models in response to global change drivers. Though this might neglect older functional type research, especially in the vegetation dynamics field.

LL162-164 and are they associated with fewer species?

I found Fig 6 a bit empty. It really just seemed a collection of pictures and some aspects of studies. ‘spatially-explicit’, experimental, correlations. I don’t find the use of triangles as indicative of number of studies very clear or compelling.

Sect 6 LL241& ff. ok, but isn’t that saying just do more of everything. Perhaps there are trade-offs in what approaches are taken. And just including more information, does not guarantee a better outcome.

LL295 & ff. (also LL272-277) I don’t get the point here. Yes, there are statistical methodologies to relate traits, environment and performance from observational data. What is the point of the last sentence? are you saying such methods don’t work? Are not enough? Or something else, in which case please be explicit about it. Are you proposing path analysis, for instance.

LL349 & ff this section would seem to benefit from some references.

Author's Response to Decision Letter for (RSPB-2021-1277.R0)

See Appendix A.

RSPB-2022-0071.R0

Review form: Reviewer 1

Recommendation

Major revision is needed (please make suggestions in comments)

Scientific importance: Is the manuscript an original and important contribution to its field?

Good

General interest: Is the paper of sufficient general interest?

Good

Quality of the paper: Is the overall quality of the paper suitable?

Good

Is the length of the paper justified?

Yes

Should the paper be seen by a specialist statistical reviewer?

No

Do you have any concerns about statistical analyses in this paper? If so, please specify them explicitly in your report.

No

It is a condition of publication that authors make their supporting data, code and materials available - either as supplementary material or hosted in an external repository. Please rate, if applicable, the supporting data on the following criteria.

Is it accessible?

Yes

Is it clear?

N/A

Is it adequate?

N/A

Do you have any ethical concerns with this paper?

No

Comments to the Author

I (reviewer 1) think the manuscript has been improved and the authors responded to the comments made to the original submission. However, I am not perfectly satisfied.

It seems the line numbers given in the response to the reviewers letter do not match the ones in the manuscript, neither the version with nor without track changes.

The point a) I do see the trade-off between further explanations and word/page limit of Proceedings B, but in my opinion, the response to that point falls too short.

d) The term "narrow assemblage" has been substituted in some cases in the ms, but it is still used without explanation. In line 110 the meaning of the term is not clear and the corresponding figure 5A is not convincingly showing, what might be meant.

Otherwise, I am satisfied with the responses.

Decision letter (RSPB-2022-0071.R0)

01-Feb-2022

Dear Dr Green

I am pleased to inform you that your manuscript RSPB-2022-0071 entitled "Trait-based approaches to global change ecology: moving from description to prediction" has been provisionally accepted for publication in Proceedings B.

The referee - who had seen the previous version -- is happy with most of your revisions, as am I, but still would like the 'narrow assemblage' issue nailed once and for all. So, there is still "Interestingly, relatively narrower assemblages of traits are used in plant compared with animal research (Figure 5A, Figure S2A&S3A)." and " Interestingly, relatively narrower assemblages of traits are used in plant compared with animal research (Figure 5A, Figure S2A&S3A)." If this really just means "fewer traits" then change it to that; likewise if it means "a lower diversity of traits". I think I understand what you've done with the nMDS. This is a 2D representation of a higher dimensional trait space - with each trait scored as present or absent in each study, studies with more traits in common will be closer in (whatever dimension of) trait space. I am guessing that the two statements above would be visualised as plant studies occupying a smaller area of trait space than animal studies. However (the referee's second issue, beyond defining the phrase more clearly) is that it's not obvious this is the case -- maybe in figure 5A, but in S2A and S3A it's hard to see a difference. Part of this is presentational - it's not clear how much of the difference in shading between polygons is due to the different categories' colours and how much is due to greater density where there is overlap. But, aside from this, most of the differences in the MCPs of S2A and S3A seem to be due to one or two points and, if anything, invertebrates seem to occupy the smallest area (but that may be because the orange points are more salient than shades of purple). How much of the differences are due to differences in numbers of studies (more studies = larger area of trait space occupied) or is that controlled for? Anyway, I agree with the referee that the link between the data analysis/presentation and the point being made about differences between taxa, needs to be clarified. Therefore, I invite you to respond to the referee's' comments and revise your manuscript. Because the schedule for publication is very tight, it is a condition of publication that you submit the revised version of your manuscript within 7 days. If you do not think you will be able to meet this date please let us know.

[http://datadryad.org/submit?journalID=RSPB&manu=\(Document not available\)](http://datadryad.org/submit?journalID=RSPB&manu=(Document%20not%20available)) which will take you to your unique entry in the Dryad repository. If you have already submitted your data to dryad you can make any necessary revisions to your dataset by following the above link. Please see <https://royalsociety.org/journals/ethics-policies/data-sharing-mining/> for more details.

Best wishes,
Innes Cuthill

Professor Innes Cuthill
Reviews Editor, Proceedings B
mailto: proceedingsb@royalsociety.org

Reviewer(s)' Comments to Author:

Referee: 1

Comments to the Author(s).

I (reviewer 1) think the manuscript has been improved and the authors responded to the comments made to the original submission. However, I am not perfectly satisfied.

It seems the line numbers given in the response to the reviewers letter do not match the ones in the manuscript, neither the version with nor without track changes.

The point a) I do see the trade-off between further explanations and word/page limit of Proceedings B, but in my opinion, the response to that point falls too short.

d) The term "narrow assemblage" has been substituted in some cases in the ms, but it is still used without explanation. In line 110 the meaning of the term is not clear and the corresponding figure 5A is not convincingly showing, what might be meant.

Otherwise, I am satisfied with the responses.

Author's Response to Decision Letter for (RSPB-2022-0071.R0)

See Appendix B.

Decision letter (RSPB-2022-0071.R1)

09-Feb-2022

Dear Dr Green

I am pleased to inform you that your manuscript entitled "Trait-based approaches to global change ecology: moving from description to prediction" has been accepted for publication in Proceedings B.

Data Accessibility section

Open Access

Paper charges

Sincerely,

Proceedings B

Appendix A

Response to reviews of "Trait-based approaches to global change ecology: moving from description to prediction" for Proceedings of the Royal Society B

Please find our detailed responses (in italics) to each reviewer's comment below. Line numbers refer to the tracked changes version of the manuscript.

Dear Dr Green:

I am writing to inform you that your manuscript RSPB-2021-1277 entitled "Trait-based approaches to global change ecology: moving from description to prediction" has, in its current form, been rejected for publication in Proceedings B.

This action has been taken on the advice of referees, who have recommended that substantial revisions are necessary. The referees do, however, differ in their view of whether the revisions recommended would take the manuscript above the acceptance threshold, with referee 1 being more positive than referee 2. While some of referee 2's negativity comes from (what the referee freely admits are) personal opinions rather than 'errors' in the manuscript, if you want your paper to have broad impact, as you surely do, then this is the sort of person you need to win over. For example, I think you should read the review by Zakharova et al. (2019) because any assessment of the value of trait-based approaches for global change ecology will gain from understanding the value of trait-based approaches in ecology per se. If you think you can make changes that would fully address the comments of both referees, I would be happy to consider a resubmission. However please note that this is not a provisional acceptance.

The resubmission will be treated as a new manuscript. However, we will approach the same reviewers if they are available and it is deemed necessary to do so. Please note that resubmissions must be submitted within six months of the date of this email. In exceptional circumstances, extensions may be possible if agreed with the Editorial Office. Manuscripts submitted after this date will be automatically rejected.

4) Please read our data sharing policies to ensure that you meet our requirements
<https://royalsociety.org/journals/authors/author-guidelines/#data>.

Best wishes,

Innes Cuthill

Prof. Innes Cuthill
Reviews Editor, Proceedings B
mailto: proceedingsb@royalsociety.org

We are grateful to the two reviewers for their detailed feedback on our manuscript and are pleased that both reviewers recognized the importance and timeliness of a synthesis of trait-based research through the lens of global change prediction. We feel that we have been able to address all the items raised by both reviewers through detailed edits and additions to the manuscript. In addition to strengthening the key points within our review, our edits also make more explicit applications and examples of the synthesis points we raise, as suggested by both reviewers. We have provided detailed information on our changes to the study below, as well as accompanying line numbers corresponding to the revised manuscript. We hope that you and the reviewers will find the revised review paper is now suitable for publication in Proceedings of the Royal Society B.

Reviewer(s)' Comments to Author:

Referee: 1

Comments to the Author(s)

Review of “Trait-based approaches to global change ecology: moving from description to prediction“ by Green et al.

This study is a literature study in the field of trait-based ecology. The authors focussed on studies dealing with global change and they were particularly interested in studies making predictions in the light of global change. This is a very timely and important task. Many trait-based studies have been published in the last decades and the time is right to get an overview and to guide research to the most urgently needed questions. A total number of 865 studies was analyzed and

a huge amount of data on traits, taxa, habitats etc. was compiled. One result was that in only very few studies trait-based approaches were used to make predictions in response to global change. I very much appreciate the approach, but I see some room for improvements.

We appreciate that the reviewer recognizes the importance and timeliness of our synthesis of trait-based literature through the lens of global change prediction. We have incorporated all of the helpful points they raise below through revisions and additions to the manuscript text. Please see below for our detailed responses and edits.

a) In my opinion, the reader would benefit from more examples of individual studies and the outcome of those studies. For example (not restricted to that) in the part from line 150 to 155. The few predictive studies are very well presented in combination with Figure 6.

We thank the reviewer for this suggestion, and also appreciate their positive feedback about our presentation of predictive studies and Figure 6. We have added explicit reference to this visual and the examples it contains in the text related to lines 150-155 (now lines 204-212 in the tracked changes version). We have also added a table summarizing the attributes of the predictive global change studies we identified in the review (new Table S5) to provide additional detailed examples while keeping within the length limits of reviews in Proceedings B. Beyond this section, we have added additional references and examples throughout the rest of the manuscript. Some examples include highlight the focus of observational research in vascular plants identifying morphological trait diversity and community-weighted mean values across abiotic gradients of light, moisture, and nutrients (lines 118-119), abiotic and dispersal filtering studies focused on plankton and fishes in aquatic systems (lines 122-123), and theoretical work in service of cross-ecosystem prediction (lines 144-146), among others. Finally, we include in our online materials the data frame of coded studies so that interested readers can more deeply review studies related to each dimension of the synthesis (accessible in GitHub, linked in the Data Availability statement).

b) I was thinking if there is a causal relationship between the majority of size as a trait (and specific leaf area) used and the low number of predictive studies. Size is relatively easy to measure, it is an important „master trait“, however, a change in size itself does not tell necessarily a lot about a change in ecosystem function; in particular if this change is small. For predictions, other and multiple traits than size might be needed.

The reviewer raises an excellent point. We have revised the text on lines 325-337 (tracked changes version) to capture the idea that information on multiple traits beyond size may often be required to explain variation or change in ecosystem function (particularly of small change/variation).

c) What was the focus of the many studies that were not making predictions? Did they simply fail or did they have a different focus? In phytoplankton, for example, it was found that in marine studies, trait-based approaches are often used to explain biogeochemical processes, whereas in freshwater studies, trait-based approaches are used to understand the community response to environmental factors/changes (Weithoff & Beisner, 2019, Front Mar Sci). Could you see similar distinctions in your data set? Would be interesting.

Studies that were not predictive primarily focused on identifying correlations between trait values with environmental. In these cases, traits are inferred to have a causal role in affecting organism response to the environmental conditions examined. We provide detail on the focus of studies that were not predictive (i.e. focused on description) but in order to keep the focus on what is known regarding forecasting global change, we regret that we are not able to provide a full description of the themes within research within particular taxa or environments within the confines of the length limits for Proceedings B reviews. Nevertheless, we have revised our description at lines 113-124 (tracked changes version) to be more specific about the focus of observational work in terrestrial plant systems (which focus on patterns of trait diversity and mean-weighted values across abiotic gradients of light, moisture, and nutrients) and work in marine and freshwater plankton and fish communities on abiotic and dispersal filtering processes with reference to example studies (lines 113-124).

d) The meaning of narrow assemblages needs more explanation. What does it tell the reader, when the trait assemblages are narrow? And what is the ecological meaning? Which traits were used in the studies that are referred to in line 99-107?

Thank you for highlighting that our language was not clear. We have clarified that by 'narrow assemblage' we mean 'fewer' or 'lower number of unique' traits represented on lines 99 and 107 (now lines 139-150 in the tracked changes version). Our ecological interpretation is that researchers have characterized a smaller set of ecological functions within these taxonomic groups and environments, which we have clarified in this section.

e) The term „novel ecosystems“ appears in the abstract and the conclusive part 7 and only once in the text. Do the authors regard any ecosystem affected by global change as a novel ecosystem? I think the term „novel ecosystems“ needs a definition when used, because for different readers, it might have different meanings. Maybe use Hobbs et al. 2009, TREE as a reference.

Thank you for this suggestion and thought-provoking question. We have a more clearly defined 'novel ecosystem', citing Hobbs et al. 2009 in this section (now lines 67-72 in the tracked changes version). In particular, we define a 'novel ecosystem configuration' as one in which species membership has changed to such an extent that core ecosystem functions are affected.

f) „Plankton“ is not a taxon. Plankton can be plants (phytoplankton) or animals (zooplankton) or something in between (mixotrophs), or procaryotes. I think it needs to be justified, why no taxonomic distinction was made.

Thank you for clarifying that plankton itself is a functional trait classification! We have clarified in Table S2 that studies in this classification focus on organisms with small body size and pelagic habitat presence within an aquatic ecosystem.

g) Line 76 I couldn't find a table displaying the journals where the studies were published in.

We apologize that this table was missing from the supplement in our original submission. We have now included Table S4 in our revision for your consideration.

h) Figure 3. Can you change the colour code so that 0 (zero) is white. At first glance it looks like quite a number of studies for all combinations.

Thank you for this suggestion. We have revised the colour code in Figure 3 so that zero is a much paler shade of yellow to address the issue while retaining the aesthetic of the figure.

Referee: 2

Comments to the Author(s)

Review of RSPB-2021-1277

This manuscript reviewed literature on studies of global change effects on organisms utilising (functional) traits. It's scope is broad, ranging across biomes and taxa as broadly as possible. The manuscript's aims are most clearly articulated in the last paragraph of Section 1. The manuscript uses a systematic review to query the literature and provide an empirical basis for questions of how are traits being used, which traits, across different fields and with respect to various processes. The review also aims to highlight beneficial ways forward to provide a predictive basis to support management decision regarding ecosystem function and community assembly in response to global change drivers.

The first question is straightforward. What does the bibliometric data tell us? The authors have comprehensively and clearly communicated the results of this enquiry.

Thank you for your positive feedback. This was indeed one of the core aims of our piece.

The next two aims I found harder to parse and to judge whether the authors had achieved them. “(2) What progress are we making in addressing potential barriers to predicting the outcome of

environmental and biotic interactions using functional traits? It is a little vague. And how is progress measured? a judgement? I could not assess whether this had been done well.

The author's feedback made us realize that we could re-word the question to be clearer. We have revised it as: "What progress are we making in addressing potential barriers to predicting the outcome of environmental and biotic interactions [under global change] using functional traits?" (lines 83-86 in tracked changes version). Our approach to this question is directly addressed in section "6. Persistent challenges and opportunities for trait-based global change prediction" (lines 303-313) In this first paragraph of this section (lines 303-312) we have clarified that our measure of progress is the presence of literature addressing previously identified gaps and needs for trait-based analyses for ecosystems. In particular, cited reviews by McGill and colleagues (2006), Violle and colleagues (2014), Enquist and colleagues (2015), and Zakharova et al. (2019) consistently highlight issues of trait number and complexity, scales of observation, and intraspecific variation. We use our synthesis to highlight literature that directly addresses these issues in the context of creating spatially-explicit process models based on traits for global change prediction.

And the 3rd, "What research methods and frameworks show promise in moving the field from description and towards more nuanced prediction of community re-assembly and function?: How is 'promise' judged?"

Your comment made us realize that it was clearer to present this question second (rather than third) in our list in the manuscript (lines 81-87 in the tracked changes version). Our approach to the question is outlined in the first paragraph of the section "5. Promising methodological frameworks for trait-based global change prediction". 'Promising' is defined as 'likely to have future success'. Based on your feedback we have clarified in the manuscript text that the three criteria we outline for choosing among forecasting methodologies (lines 214-228) are our criteria for likely future success in predicting community re-assembly and function under global change. In particular, we have revised this section as: "...whether the method: 1) explicitly integrates mechanistic knowledge or hypotheses about the environmental filter(s) likely to be disrupted by focal global change driver[s]; Figure 1), 2) considers scales (taxonomic, spatial, and temporal) at which functional trait and environmental condition information are both required and available for the focal system, and 3) the forecasting output (e.g. point estimate or probability distribution, threshold value or limit, spatially or temporally explicit) is relevant for the specific conservation or management interventions being considered in the system to address the effects of the global change driver in question."

But those quibbles with wording of aims aside, I am not clear how to judge this work. While the manuscript's breadth is a strength, it perhaps is the study's weakness, because it leads to some broad statements that come across as quite vague. On the one hand I am sympathetic to the aim

of using traits in models, to be predictive. And I see the merit of reviewing the state of play. And yet I feel that probably considerable predictive modelling is missed by this review, because of the breadth addressed. It also has resulted in a lack of specificity, in my opinion.

We appreciate that the reviewer sees the value of quantifying the status of traits-based efforts to predict global change outcomes. Given the quantitative nature of our review, it is certainly important that we properly characterize the state of trait-based predictive literature; Indeed, casting an appropriately wide net was top of mind as we developed and tested our literature review approach (described in Table S1). The field of predictive ecological modeling is vast. Our intent was not to review predictive ecological studies in general, but rather those that explicitly 1) self-identify as focusing on traits-based classifications of organisms within their focal community, and within this subset of studies, are 2) focused on global change drivers. When existing work is further filtered by these two criteria, the body of work shrinks substantially. This was also surprising to us, and the major reason for conducting this review—to highlight what is known, but most importantly what is not.

The reviewer's comments made us realize that it would be helpful to define more clearly what we mean by 'predictive ecological model' in the context of our study. In our case, we classified studies as predictive when they generated model output for beyond the data set for which parameter values were initially estimated. Included within this definition are theoretical and process-oriented models that predict outcomes based on parameter inputs generated from initial values, and forecasts generated from statistical models that use data other than the information on which the initial model was trained for forecasting. We have clarified this definition in Table S2 which provides the detailed criteria for our classification scheme. We see prediction (i.e. forecasting) as a distinct step that follows ecological model building; i.e. building an ecological model is a prerequisite for forecasting, but not all ecological model studies complete this next step.

It is my belief that traits are useful in a comparative context to assess the capacity for, and to facilitate, generalisation. This aspect appears to be ignored or at least not highlighted. Instead, the highlight is placed on using traits in (modelling effects of) global change.....

We see the use of traits for modeling the effects of global change and in a comparative context to facilitate generalization (as the reviewer states) as purposes that are highly complementary; Indeed, identifying generalizable traits through comparative work is a prerequisite for subsequently using these traits to predict outcomes of global change forcing for ecosystems. Given the rapid pace at which change is occurring, we feel this is the most urgent application of generalizable trait-based knowledge today. We have revised the text at the outset of our manuscript on lines 38-41 (tracked changes version) to emphasize the importance of comparative work in identifying generalizable traits, as well as the need for researchers to focus comparative work on traits that are most likely to drive processes that are being disrupted.

...Now, I agree with that research program of using traits to predict processes and outcomes. And I agree entirely with the need to match scales of investigation (drivers and responses) and the outcomes of importance to managers. However, broad-scale exploratory studies (correlations) enable the basis for fine-scale investigations and predictions to be made. To dismiss correlative descriptive studies of traits seems to me to miss the point.

We agree with the reviewer's point that correlative studies are a key source of information on the identity of generalizable traits that are likely to be driving key ecological processes and outcomes. We acknowledge the important role of observational data on lines 241-244 in the tracked changes version ("Data generated from experiments and observational methods are essential inputs into modeling frameworks (Figure 6; Table S5)...."). Nevertheless, the reviewer's comment made us realize that we need to better ensure that readers do not conflate our focus on prediction with dismissal of observational studies. We therefore have edited language at the outset of our manuscript on lines 36-41 to clarify the integral role of observation in identifying generalizable traits that drive ecological processes and thus in facilitating predictions about ecological outcomes in new contexts.

By contrast, in at least one of the papers highlighted as a good example of using traits to predict change, no mention is made of where those parameters came from (ref 28. Fordham et al 2012). Often demographic models are parameterised with best guesses and observations (of demographic rates), the promise of traits would be to provide estimates of those rates from measurable traits (e.g. extending the exploratory study of Adler et al 2014 <https://doi.org/10.1073/pnas.1315179111>).

The reviewer raises a good point about the synthetic nature of process models and the opportunity to inform them with measured trait estimates derived from multiple sources. We have explicitly added this point to lines 242-245 in the manuscript (tracked changes version).

While specifying traits, and then accepting a very wide range of traits (eg habitat type, competitive ability (specified how?)), the authors appeared to not include functional groups. Which is curious because much (predictive) global change modelling (e.g. dynamic global vegetation dynamics modelling) has used functional type approaches. So I would expect some discussion of how functional groups or types have been used in modelling and why traits should be considered different or better from them and with specific reference to the purpose and constraints of predictive modelling. My own feeling (only opinion) is that specifying types offers convenience for modelling and its communication, relative to continuous traits.

Thank you for suggesting that we highlight categorical functional feeding groups in the manuscript text, which were a large focus of many of the studies we reviewed. Often such groups

are formed by the qualitative aggregation of multiple traits. We have modified and added to the text on lines 344-355 (tracked changes version) to briefly describe trade-offs between measuring individual traits using continuous measures (which can make for more robust statistical analysis, but be more time/energy intensive to gather) and classifying categorical synthetic ‘functions’ (which may be simple to assess and apply, particularly in the context of conservation planning, but challenging to analyse and reproduce).

I found it interesting that a recent review (2019) of trait-based modelling was not mentioned (Zhakarova <https://doi.org/10.1016/j.ecolmodel.2019.05.008>) even though the topics overlap considerably. In my opinion that paper, focussed on modelling rather than global change, is more explicit about what has been done and why and how to proceed.

Thank you for highlighting this paper. The reviewer’s comments make us realize that we need to both ensure that the paper is explicitly cited in our manuscript in the appropriate section and clarify how our piece differs from this other review. As the reviewer points out, Zhakarova et al is focused specifically on modeling studies (though not explicitly stated, the review focuses primarily on process-based models), and thus reviews and highlights a subset of existing trait-based ecological research captured within our review. In contrast, we capture a range of other methodological approaches (experiments, observation, meta-analyses, and models) beyond modeling in our synthesis. Our review casts a larger net to assess the state of trait-based research as it relates to both trait-based descriptions AND predictions of ecosystems, and subsequently filter down to global change prediction. We feel this approach was necessary for assessing the extent to which a range of trait-based methods have been applied to global change questions. Nevertheless, as the reviewer points out, this review is extremely relevant. We now cite the paper throughout our manuscript (reference [15]), including within the section “Promising methods and approaches for moving from description to prediction” (Section 6); In particular, Zakharova et al.’s focus on the need to better incorporate intraspecific trait variation, compare trait and taxonomic models, and incorporate traits into individual-based modeling (a prominent technique for process-based modeling) is very relevant.

Specific minor comments

Abstract What does it mean to “propagate environmental (and trait) variation throughout process model functions”?

We have clarified that we mean “propagate variation in trait and environmental parameters throughout process model functions through simulation” [...as opposed to using point estimates/single trait values].

L57&ff how did you deal with line cases? morphological vs physiological?

We're unclear what the reviewer means by "line cases". The manuscript text at this line refers to the total number of studies analyzed and our classification scheme. If they are referring to papers in which multiple trait types were studied (i.e. both morphological and physiological), such a paper would be included in our count for both trait types.

L76 this does not match the Supplement Table S4

The sample size cited in the supplemental table (now S7) is smaller than the total sample size of N=865 papers reviewed because traits in some studies did not have a secondary classification and thus were excluded from the analysis. We have clarified this in the captions for the supplementary figures giving the mvGLM results.

L93 SLA is not a measure of plant size. it is a measure of the leaf area deployed for carbon gain, per unit of mass invested. A very different measure. It could be thought of as an 'efficiency'. Plant size is often measured by height or lateral spread or stem diameter or biomass. Leaf Area might be thought of a measure of size, within a restricted domain of plants, especially herbs and grasses, and mostly within species rather than between species.

Thank you for clarifying this point. We have replaced 'size' with 'growth efficiency'.

LL150 & ff this is the nub. Very few trait-based predictive models in response to global change drivers. Though this might neglect older functional type research, especially in the vegetation dynamics field.

Yes, we agree; there is lots of room for growth in this field. We appreciate your comment about the use of the term 'functional type'. A Google Scholar search for 'functional type' prior to 1990 yielded results that predominantly included the term 'trait' (the focus of our search).

LL162-164 and are they associated with fewer species?

We're not clear what the reviewer is asking to be compared. The manuscript text cited is "In general, physiological traits related to resource acquisition and requirements, such as thermal tolerance, and life history traits are more often applied within predictive studies compared with descriptive work (Figure 6, Figures S6&S7, Tables S10-13)". We would be happy to address this concern if the reviewer can provide additional context for their comment.

I found Fig 6 a bit empty. It really just seemed a collection of pictures and some aspects of studies. 'spatially-explicit', experimental, correlations. I don't find the use of triangles as indicative of number of studies very clear or compelling.

We have revised the figure to provide a count of studies using each type of methodology and re-scaled the triangles accordingly. We note that Reviewer 1 found Figure 6 and its accompanying text to be useful and compelling way to summarize key information about the types of studies identified by this review as well provide explicit examples of predictive traits-based global change studies. Given that both reviewers have indicated that illustrative examples are important to support the quantitative results of the literature review, our paper provides, we have elected to retain this figure in a modified form.

Sect 6 LL241& ff. ok, but isn't that saying just do more of everything. Perhaps there are trade-offs in what approaches are taken. And just including more information, does not guarantee a better outcome.

We believe the reviewer is referring to the three challenges we identify for advancing methods for traits-based global change prediction ("Realizing the full potential frameworks...."). The reviewers comment makes us realize that our purpose in highlighting these three research areas (which have been identified as key focal areas for advancing trait based analyses in multiple reviews, and are thus thorny, persistent issues) is to leverage our quantitative review to comment on their status (which has not been done in previous work), and connect these research areas back to our recommended framework (spatially explicit process modeling). We have revised our text on lines 277-287 (tracked changes version) to make this purpose explicit.

LL295 & ff. (also LL272-277) I don't get the point here. Yes, there are statistical methodologies to relate traits, environment and performance from observational data. What is the point of the last sentence? are you saying such methods don't work? Are not enough? Or something else, in which case please be explicit about it. Are you proposing path analysis, for instance.

Thank you for pointing out that we could have been clearer in this section. We agree! We aim to make three points in this section. First, that matching the scale(s) of trait and environmental data collection is important for prediction. Second, that many of the observational studies we reviewed do not match the scales of these data. Third, that the relevant scale can be selected via two methods; i) a priori, based on information or hypotheses about the mechanisms by which traits mediate the environmental filtering process, or ii) post-hoc through multivariate analyses of trait, environment, and abundance/distribution data (e.g. 4th corner or RLQ). We have revised this section accordingly, which includes removing the last sentence that was initially confusing.

LL349 & ff this section would seem to benefit from some references.

Your comment made us realize that this section could be made clearer and more concise by deleting these two sentences, which repeat information and points made elsewhere in the review (but without citation). We feel this change has made for a stronger ending. Thank you.

Appendix B

Response to reviews of the revised manuscript "Trait-based approaches to global change ecology: moving from description to prediction" for Proceedings of the Royal Society B

Please find the editor and reviewer's comments and our responses (*italics*) below.

Editor's comments:

Dear Dr Green

I am pleased to inform you that your manuscript RSPB-2022-0071 entitled "Trait-based approaches to global change ecology: moving from description to prediction" has been provisionally accepted for publication in Proceedings B.

The referee - who had seen the previous version -- is happy with most of your revisions, as am I, but still would like the 'narrow assemblage' issue nailed once and for all. So, there is still "Interestingly, relatively narrower assemblages of traits are used in plant compared with animal research (Figure 5A, Figure S2A&S3A)." and " Interestingly, relatively narrower assemblages of traits are used in plant compared with animal research (Figure 5A, Figure S2A&S3A)." If this really just means "fewer traits" then change it to that; likewise if it means "a lower diversity of traits". I think I understand what you've done with the nMDS. This is a 2D representation of a higher dimensional trait space - with each trait scored as present or absent in each study, studies with more traits in common will be closer in (whatever dimension of) trait space. I am guessing that the two statements above would be visualised as plant studies occupying a smaller area of trait space than animal studies. However (the referee's second issue, beyond defining the phrase more clearly) is that it's not obvious this is the case -- maybe in figure 5A, but in S2A and S3A it's hard to see a difference. Part of this is presentational - it's not clear how much of the difference in shading between polygons is due to the different categories' colours and how much is due to greater density where there is overlap. But, aside from this, most of the differences in the MCPs of S2A and S3A seem to be due to one or two points and, if anything, invertebrates seem to occupy the smallest area (but that may be because the orange points are more salient than shades of purple). How much of the differences are due to differences in numbers of studies (more studies = larger area of trait space occupied) or is that controlled for? Anyway, I agree with the referee that the link between the data analysis/presentation and the point being made about differences between taxa, needs to be clarified. Therefore, I invite you to respond to the referee's comments and revise your manuscript. Because the schedule for publication is very tight, it is a condition of publication that you submit the revised version of your manuscript within 7 days. If you do not think you will be able to meet this date please let us know.

We are very pleased that you and the reviewer felt that the revised manuscript was improved and are excited to see the work published in Proceedings B. We appreciate you and the reviewer pressing on our description of trait diversity and composition within the studies. After reviewing your explanation, we see that there is certainly room for us to be clearer and more precise about the results we are contextualizing. We have revised the text at lines 79, 90-98, and 109-122 (clean version) to make better use of the supplementary figures S5 through S15, which are the rank abundance plots visualizing both the total number of unique traits in each group of studies as well as the most common traits. We have also revised our language related to the nMDS, which show the similarity of studies within a particular domain in terms of the traits they use. We acknowledge that the difference between the size of the convex hulls in Figure 5a (nMDS plots) is small, and that the effect disappears when considering primary trait classifications because most major types of traits are used across taxonomic domains. However, one might expect the trait space (i.e., convex size) for plants to be much larger than the other categories given the large number of studies and unique traits used in plant research. Thus, seeing the opposite effect is a bit counter intuitive and suggests to us that perhaps that some dominant traits are used across much of this work (as per our proposed mechanisms described on lines 99-108). We have also made minor revisions to the caption of Figure 5 (the nMDS plot) and our description of the nMDS methods in the Supplementary Materials. Please let us know if these text revisions satisfactorily address this concern.

Reviewer(s)' Comments to Author:

Referee: 1

Comments to the Author(s).

I (reviewer 1) think the manuscript has been improved and the authors responded to the comments made to the original submission. However, I am not perfectly satisfied.

It seems the line numbers given in the response to the reviewers letter do not match the ones in the manuscript, neither the version with nor without track changes.

The point a) I do see the trade-off between further explanations and word/page limit of Proceedings B, but in my opinion, the response to that point falls too short.

Our revised manuscript included several additional descriptions of specific studies and a new table providing in-depth information for the trait-based global change studies we reviewed. We are unsure what specifically the reviewer would like to see included in the manuscript (and for which section(s)) to further address this issue within the limits of the Proceedings B manuscript parameters. We would be happy to include additional studies if the reviewer has specific suggestions for where they feel this is needed and where we might remove material to balance the addition.

d) The term "narrow assemblage" has been substituted in some cases in the ms, but it is still used

without explanation. In line 110 the meaning of the term is not clear and the corresponding figure 5A is not convincingly showing, what might be meant. Otherwise, I am satisfied with the responses.

We appreciate the reviewer pressing on our description of trait diversity and composition within the studies and see that there is certainly room for us to be clearer and more precise about the results we are contextualizing. We have revised the text at lines 79, 90-98, and 109-122 (clean version) to make better use of the supplementary figures S4-S15, which are the rank abundance plots visualizing both the total number of unique traits in each group of studies as well as the most common traits. We have also revised our language related to the nMDS, which show the similarity of studies within a particular domain in terms of the traits they use. We acknowledge that the difference between the size of the convex hulls in Figure 5a (nMDS plots) is small, and that the effect disappears when considering primary trait classifications because most major types of traits are used across taxonomic domains. However, one might expect the trait space (i.e. convex size) for plants to be much larger than the other categories given the large number of studies and unique traits used in plant research. Thus seeing the opposite effect is a bit counter intuitive, and suggests to us that perhaps that some dominant traits are used across much of this work (as per our proposed mechanisms described on lines 99-108). We have also made minor revisions to the caption of Figure 5 (the nMDS plot) and our description of the nMDS methods in the Supplementary Materials. Please let us know if these text revisions satisfactorily address this concern.